# DISCRIMINATIVE ACTIVE LEARNING

## ABSTRACT

We propose a new batch mode active learning algorithm designed for neural networks and large query batch sizes. The method, Discriminative Active Learning (DAL), poses active learning as a binary classification task, attempting to choose examples to label in such a way as to make the labeled set and the unlabeled pool indistinguishable. Experimenting on image classification tasks, we empirically show our method to be on par with state of the art methods in medium and large query batch sizes, while being simple to implement and also extend to other domains besides classification tasks. Our experiments also show that none of the state of the art methods of today are clearly better than uncertainty sampling, negating some of the reported results in the recent literature.

## 1 INTRODUCTION

In the past few years deep neural networks achieved groundbreaking results in many machine learning tasks, ranging from computer vision to speech recognition and NLP. While these successes can be attributed to the increase in computational power and to the expressiveness of neural networks, they are also a clear result of the use of massive datasets which contain millions of samples. The collection and labeling of such datasets is highly time consuming and often expensive, and much research has been done to try and reduce the cost of this process.

Active learning is one field of research which tries to address the difficulties of data labeling. It assumes the collection of data is relatively easy but the labeling process is costly, which is the case in many language, vision and speech recognition tasks. It tackles the question of which samples, if labeled, will result in the highest improvement in test accuracy under a fixed labeling budget. The most popular active learning setting is pool-based sampling (Settles, 2012), where we assume that we have a small labeled dataset $\mathcal{L}$ and access to a large unlabeled dataset $\mathcal{U}$ from which we choose the samples to be labeled next. We then start an iterative process where in every step the active learning algorithm uses information in $\mathcal{L}$ and $\mathcal{U}$ to choose the best $x \in \mathcal{U}$ to be labeled. $x$ is then labeled by an oracle and added to $\mathcal{L}$, and the process continues until we have reached the desired amount of samples or test accuracy.

Many heuristic methods have been developed over the years (Settles, 2012), mostly for classification tasks, that have been shown to reduce the sample size by up to an order of magnitude. These methods fit a model to the labeled dataset and then choose a single sample to be labeled according to the performance of the model on the samples in $\mathcal{U}$. While these methods are very successful, they have a few shortcomings that make them less suited for large scale data collection for deep learning. First, it is highly impractical to query a single sample from $\mathcal{U}$ in every iteration, since we need a human in the loop for every one of those iterations. Adding to that the time needed to fit a neural network, collecting a large dataset using this setting simply takes too long. Second, many of these methods are designed solely for classification tasks, and extending them to other domains is often non trivial.

To tackle the first problem, the batch-mode active learning setting was introduced, allowing for the querying of a batch of examples at every iteration. This change adds complexity to the problem since it might not be enough to greedily choose the top-$K$ samples from $\mathcal{U}$ according to some score, since they may be very similar to each other. Choosing a batch of diverse examples is often better than choosing one containing very similar examples, which may lead to performance that is worse than random sampling. Several methods were developed with this problem in mind (Chen & Krause, 2013), (Guo & Schuurmans, 2008), (Azimi et al., 2012). The second problem however has not been

explicitly addressed to the best of our knowledge, and active learning solutions to other domains have had to be adapted separately (Settles, 2008), (Vezhnevets et al., 2012).

To address the second problem and make the method agnostic to the specific task being performed, we do not assume access to a classifier trained on the labeled set when choosing the batch to query and instead only assume access to a learned representation of the data, $\Psi(x)$. We pose the active learning problem as a simple binary classification problem and choose the batch to query according to a solution to said problem. We compare our method empirically to today's state-of-the-art active learning methods for image classification, showing competitive results while being simple to implement and conceptually simple to transfer to new tasks.

## 2 RELATED WORK

There has been extensive work on active learning prior to the advances in deep learning of the recent years. This work is summarized in (Settles, 2012) which covers the many different heuristics developed for active learning, along with adaptation made for other tasks besides classification. A large empirical study of different heuristics applied to classification using the logistic regression model can be found in (Yang & Loog, 2018). This study shows uncertainty-based methods (Lewis & Gale, 1994) perform well on a large and diverse set of datasets while being very simple to implement. These methods score the unlabeled examples according to the classifier's uncertainty about their class, and choose the example with highest uncertainty.

While neural networks for classification tasks output a probability distribution over labels, they have been shown to be over-confident and their softmax scores are unreliable measures of confidence. Recent work (Gal et al., 2017) shows an equivalence between dropout, a widely used neural network regularization method, and approximate Bayesian inference. This allows for the use of dropout at test time in order to achieve better confidence estimations of the network's outputs. Our empirical analysis shows this method performs very similarly to uncertainty sampling using the regular softmax scores.

In addition to uncertainty-based approaches to active learning, another approach that has been actively studied in the past is the margin-based approach, which has a similar flavor. Instead of scoring an example based on the output probabilities of the model, these methods choose the examples which are closest to the decision boundary of the model. While in linear models like logistic regression for binary classification problems these two approaches are equivalent, this isn't the case for neural networks that have an intractable decision boundary. (Ducoffe & Precioso, 2018) approximates the distance to the decision boundary using the distance to the nearest adversarial example (Szegedy et al., 2013). Our empirical analysis shows that while this method is conceptually very different from uncertainty sampling, it ranks the unlabeled examples in a very similar way.

Another active learning approach is based on expected model change, choosing examples which will cause the biggest effect to our model when labeled. (Settles et al., 2008) and (Huang et al., 2016) use this approach by calculating the expected gradient length (EGL) of the loss with respect to the model parameters, when the expectation is over the posterior distribution over the labels induced by the trained classifier. While their results show significant improvements over random sampling for text and speech recognition tasks, our empirical analysis along with the analysis of (Ducoffe & Precioso, 2018) show that this method is not suited for image classification using CNNs.

Finally, (Sener & Savarese, 2018) and (Geifman & El-Yaniv, 2017) address the challenge posed by the batch-mode setting by formulating the active learning task as a core set selection problem, attempting to choose a batch which $\epsilon$-covers the unlabeled set as closely as possible and obtains a diverse query in the process. This is done robustly by formulating the $\epsilon$-cover as an integer program which allows a small subset of the examples (outliers) not to be covered. The metric space over which the $\epsilon$-cover is optimized is the learned representation of the data. While not discussed in their paper, the core set approach is similar to our method in that it only requires access to a learned representation of the data and not a trained classifier, which allows their method to easily extend to other tasks as well. However, their robust formulation is not as straightforward to implement as our method and does not scale as easily to large unlabeled datasets.

# 3 DAL - Discriminative Active Learning

We motivate our proposed method with a simple idea - when building a labeled dataset out of a large pool of unlabeled examples, we would like our dataset to represent the true distribution of the data as well as possible. In high dimensional and multi-modal data, this would ensure our labeled set has samples from all of the different modalities of the distribution. A naive solution would be to model the distribution of the unlabeled pool and use the learned distribution to generate a sub-sample for labeling, but the generative models of today are either hard to train or break down in high dimensional problems, so a different solution is needed.

Assuming the unlabeled pool is large enough to represent the true distribution, we can instead ask for every example **how certain we are that it came from the unlabeled pool as opposed to our current labeled set**. If the examples from the labeled set are indistinguishable from the unlabeled pool, then we have successfully represented the distribution with our labeled set. Another intuitive motivation is that if we can say with high probability that an unlabeled example came from the unlabeled set, then it is different from our current labeled examples and labeling it should be informative.

This proposition allows us to pose the active learning problem as a binary classification problem between the unlabeled class, $\mathcal{U}$, and the labeled class, $\mathcal{L}$. Given a simple binary classification task, we can now leverage powerful modern classification models to solve our problem.

Formally, with $\Psi : \mathcal{X} \to \hat{\mathcal{X}}$ being the mapping from the original input space to the learned representation, we define a binary classification problem with $\hat{\mathcal{X}}$ as our input space and $\mathcal{Y} = \{l, u\}$ as our label space, where $l$ is the label for a sample being in the labeled set and $u$ is the label for the unlabeled set. For every iteration of the active learning process, we solve the classification problem over $\mathcal{U} \cup \mathcal{L}$ by minimize the log loss and obtaining a model $\hat{P}(y|\Psi(x))$. We then select the top-$K$ samples that satisfy $\operatorname*{argmax}_{x \in \mathcal{U}} \hat{P}(y = u|\Psi(x))$.

## 3.1 Relation to Domain Adaptation

We next show a connection to domain adaptation in the binary classification setting. This connection gives some theoretical motivation to our method.

Domain adaptation deals with the problem of bounding the error of a classifier trained on a source distribution $\mathcal{D}_S$ but then tested on a different target distribution $\mathcal{D}_T$. Naturally, we are interested in how different the two distributions are. Following (Kifer et al., 2004), (Ben-David et al., 2010) suggests the $\mathcal{H}$-divergence as a relevant distance:

**Definition 3.1** *Given a domain $\mathcal{X}$ and two distributions, $\mathcal{D}_S$ and $\mathcal{D}_T$, over $\mathcal{X}$, and given a hypothesis class $\mathcal{H}$ over $\mathcal{X}$, the $\mathcal{H}$-divergence between $\mathcal{D}_S$ and $\mathcal{D}_T$ is*

$$d_{\mathcal{H}}(\mathcal{D}_S, \mathcal{D}_T) = 2 \sup_{h \in \mathcal{H}} |\mathbb{P}_{x \sim \mathcal{D}_S}[h(x) = 1] - \mathbb{P}_{x \sim \mathcal{D}_T}[h(x) = 1]|$$

Furthermore, (Ben-David et al., 2010) relates the domain adaptation error to $d_{\mathcal{H}}(\mathcal{D}_S, \mathcal{D}_T)$, showing that the test error over $\mathcal{D}_T$ of a classifier which was trained on $\mathcal{D}_S$ is bounded by a term that depends on $d_{\mathcal{H}}(\mathcal{D}_S, \mathcal{D}_T)$.

The similarity to our case is straightforward: we train a predictor based on the labeled examples, and expect it to also work well on the unlabeled examples. To do so, we would like the distributions over the labeled and unlabeled examples to be as similar as possible.

Formally, we will construct a source and target distributions as follows: the "target" distribution will be the distribution which is focused evenly on the examples in the unlabeled set, namely, $\mathcal{D}_T = \frac{1}{|\mathcal{U}|} \sum_{x \in \mathcal{U}} \delta(x)$. Similarly, the "source" distribution will be $D_S = \frac{1}{|\mathcal{L}|} \sum_{x \in \mathcal{L}} \delta(x)$. Observe that the $\mathcal{H}$ divergence becomes

$$d_{\mathcal{H}}(\mathcal{D}_S, \mathcal{D}_T) = 2 \sup_{h \in \mathcal{H}} \left| \frac{1}{|\mathcal{L}|} \sum_{x \in \mathcal{L}} h(x) - \frac{1}{|\mathcal{U}|} \sum_{x \in \mathcal{U}} h(x) \right|$$

Solving this problem is equivalent to finding a classifier in $\mathcal{H}$ that discriminates between $\mathcal{L}$ and $\mathcal{U}$. So, if we want to minimize the difference in performance between source and target distributions, we should minimize $d_{\mathcal{H}}(\mathcal{D}_S, \mathcal{D}_T)$. DAL can be viewed as a proxy to minimizing $d_{\mathcal{H}}(\mathcal{D}_S, \mathcal{D}_T)$ in a greedy manner — at each iteration, we first (approximately) find the $h$ that achieves the supremum in the definition of $d_{\mathcal{H}}(\mathcal{D}_S, \mathcal{D}_T)$ and then we move an example from $\mathcal{U}$ to $\mathcal{L}$ so as to decrease $d_{\mathcal{H}}(\mathcal{D}_S, \mathcal{D}_T)$.

### 3.1.1 CONNECTION TO DOMAIN ADAPTATION BY BACKPROPAGATION

It is interesting to compare our method to the unsupervised domain adaptation method introduced in (Ganin & Lempitsky, 2014). Their method aims to learn a neural network that will generalize well to a target domain by demanding the representation both minimize the training loss on the source domain and maximize the binary classification loss of a classifier that distinguishes between the source and target representations. Here, the classification and domain adaptation objectives are trained jointly, while in DAL they are decoupled - the representation is learned by optimizing the classification objective and the sample to query is then chosen using the domain adaptation objective given the learned representation.

It would be interesting to use this domain adaptation framework in an active learning setting, as using the unlabeled pool in addition to the labeled set to learn the representation could be beneficial. This can be seen as an extension of DAL, which we leave as future work.

## 3.2 RELATION TO GANs

Our method is reminiscent of generative adversarial networks (Goodfellow et al., 2014) in that it attempts to fool a discriminator which tries to distinguish between data coming from two different distributions. However, while the distribution created by the generator in GANs is differentiable and changes according to the gradient of the discriminator in an attempt to maximize the discriminator loss, the active learning setting does not allow us to change the distribution of $\mathcal{L}$ in a differentiable way. Rather, we may only change $\mathcal{L}$ by transferring examples from $\mathcal{U}$, which is a discrete and non differentiable decision.

In that sense, DAL's choice of example being the one which the discriminator is most confident came from $\mathcal{U}$ can be seen as a greedy maximization of the loss of the discriminator, extending the GAN objective to a non differentiable setting. Instead of modeling the distribution of $\mathcal{U}$ with a neural network (the generator), we model the distribution using $\mathcal{L}$ which is a subset of $\mathcal{U}$. Since we train the discriminator until convergence before choosing the examples to label, it gets to observe all of the examples of $\mathcal{L}$ and $\mathcal{U}$ and more easily detect modes which exist only in $\mathcal{U}$. This aids in avoiding the problem of mode collapse often exhibited by GANs.

It should be noted that there have been direct attempts to use GANs for active learning (Zhu & Bento, 2017) in the membership synthesis active learning setting (Settles, 2012), but the results so far have been preliminary and do not improve upon random sampling when evaluating the method on real datasets. Since so far GANs have been hard to train and unstable when working on real data, our method allows a simpler way of tackling the objective of GANs for active learning.

## 3.3 DIFFERENCE FROM THE CORE SET APPROACH

It should be noted that similarly to the core set approach, our approach eventually covers the unlabeled data manifold with labeled examples, while not explicitly optimizing an $\epsilon$-cover objective. This could suggest a similarity between the two methods, but they differ on data manifolds which aren't uniform in their density.

The core set approach attempts to cover all of the points in the manifold without considering density, which leads it to over represent examples which come from sparse regions of the manifold. On the other hand, assuming DAL is able to represent $\hat{P}(y|\Psi(x))$ correctly, it will choose examples such that the ratio of labeled to unlabeled examples remains the same over the entire manifold. This means that $\mathcal{L}$ will not be biased towards sparse regions of the manifold, and will sample from $\mathcal{U}$ in proportion to the density. Experiments comparing the two methods on synthetic data confirm this intuition. However, this does not guarantee our method should perform better than the core

---

**Algorithm 1** Discriminative Active Learning

---

1: **procedure** DAL QUERY($\mathcal{U}, \mathcal{L}, K, n$)  ▷ $K$ is the total budget, $n$ is the amount of mini-queries
2:   **for** i=1...n **do**
3:     $P \leftarrow TRAIN\_BINARY\_CLASSIFIER(\mathcal{U}, \mathcal{L})$
4:     **for** j=1...$\frac{K}{n}$ **do**
5:       $\hat{x} \leftarrow \underset{x \in \mathcal{U}}{\arg\max}\, P(y = u | \Psi(x))$
6:       $\mathcal{L} \leftarrow \mathcal{L} \cup \hat{x}$
7:       $\mathcal{U} \leftarrow \mathcal{U} \backslash \hat{x}$
8:     **end for**
9:   **end for**
10:   **return** $\mathcal{U}, \mathcal{L}$
11: **end procedure**

---

set approach, since a bias towards sparse regions of the manifold may be advantageous in certain situations.

## 4 PRACTICAL CONSIDERATIONS

We move to detailing some of the practical considerations that arise when implementing DAL.

### 4.1 TRADING OFF SPEED FOR QUERY DIVERSITY

As defined above, our method is greedy and chooses the top-$K$ examples to be labeled according to our score function, without consideration for the diversity of the batch. Observe however that if we keep $\Psi$ fixed, **we do not need a human to label the chosen examples in order to move an example from $\mathcal{U}$ to $\mathcal{L}$ and continue optimizing**. The representation of the chosen example stays the same and all we need to do is mark it as labeled. If the sample size we want to query is $K$, we can split the query into $n$ mini-queries by first choosing only the top-$\frac{K}{n}$ examples to be labeled using our method, moving those examples to $\mathcal{L}$ without actually labeling them yet, training a new classifier on the new $\mathcal{L}$ and repeating the process. This ensures the eventual chosen query is diverse, since consecutive mini queries will be less likely to contain similar instances. $n$, the amount of times we split the query and repeat this process, gives us a parameter with which to trade-off between the running time of our algorithm and the amount of awareness it has for the diversity of its query. The final algorithm is detailed in 1. In practice, we see an improvement in performance when using these kind of mini queries, but after a certain number of mini queries the improvement levels off. For the datasets used in these experiments, we found 10-20 mini queries to perform well.

### 4.2 CHOOSING THE ARCHITECTURE FOR THE BINARY CLASSIFICATION

Since the learned representation isn't structured in a way that should give a convolutional architecture any advantage, we chose to use a multi-layered perceptron as the architecture for solving the binary classification task. The chosen architecture was arbitrary, having three hidden layers with a width of 256 and ReLU activation functions.

### 4.3 HANDLING OVERFITTING ISSUES

Note that we are not solving the binary classification problem with generalization in mind, but rather we are trying to fit the training set as well as possible and detect the examples which are harder to fit. A large neural network is often expressive enough to achieve a loss of zero on this kind of binary training set, but it would find it harder to fit samples which are nearby and of different labels. This means that using a small neural network or using early stopping can ensure that the samples which don't have a high probability of coming from $\mathcal{U}$ will have a lower $\hat{P}(y = u | \Psi(x))$. In practice, we found that for an unlabeled pool 10 times the size of the labeled set, stopping the training process when the training accuracy reached ~98% performed well, but changing this value slightly didn't have a significant effect on performance.

## 5 EMPIRICAL EVALUATION

### 5.1 DATASETS AND MODELS

We evaluated our method on two datasets, MNIST (LeCun, 1998) and CIFAR-10 (Krizhevsky & Hinton, 2009). We used the same model to evaluate the labeled dataset of every method. For the MNIST task our classification model was a simple LeNet architecture (LeCun et al., 1998), while for the CIFAR-10 task we used VGG-16 (Simonyan & Zisserman, 2014) as our architecture.

### 5.2 EXPERIMENTAL SETUP

There isn't a standard methodology for evaluating active learning algorithms with neural networks today. In reviewing the literature, an important factor which changes between algorithm evaluations is the query batch size, ranging from dozens of examples all the way up to thousands. Since the query batch size is expected to have a significant effect on the performance of the different methods, we evaluate the methods with query batch sizes of different orders of magnitude.

A single experiment for an algorithm entailed starting with an initial random batch which is identical for all algorithms. We then select a batch to query using the algorithm, add it to the labeled dataset and repeat this process for a predetermined number of times. In every intermediate step we evaluate the current labeled dataset by training the model on the dataset for many epochs and saving the weights of the epoch with the best validation accuracy, where a random 20% of the labeled set is reserved as the validation set. The test accuracy of the best model is then recorded. To reduce the variance from the stochastic training process of neural networks, we average our results over many experiments for every algorithm. We optimize the model using the Adam optimizer (Kingma & Ba, 2014) with the default learning rate of 0.001.

The complete code used for these experiments is available online (Anonymous, 2018).

### 5.3 EVALUATED ALGORITHMS

We compare our method to the following baselines:

- **Random**: the query batch is chosen uniformly at random out of the unlabeled set.
- **Uncertainty**: the batch is chosen according to the uncertainty sampling decision rule. We report the best performance out of the max-entropy and variation-ratio acquisition functions.
- **DBAL** (Deep Bayesian Active Learning): according to (Gal et al., 2017), we improve the uncertainty measures of our neural network with Monte Carlo dropout and report the best performance out of both the max-entropy and variation-ratio acquisition functions.
- **DFAL** (DeepFool Active Learning): according to (Ducoffe & Precioso, 2018), we choose the examples with the closest adversarial example. We used cleverhans' (Papernot et al., 2018) implementation of the DeepFool attack (Moosavi-Dezfooli et al., 2016).
- **EGL** (Expected Gradient Length): according to (Huang et al., 2016), we choose the examples with the largest expected gradient norm, with the expectation over the posterior distribution of labels for the example according to the trained model.
- **Core-Set**: we choose the examples that best cover the dataset in the learned representation space. To approximate the $\epsilon$-cover objective, we use the greedy farthest-first traversal algorithm and use it to initialize the IP formulation detailed in (Sener & Savarese, 2018). We use Gurobi (Gurobi Optimization, 2018) to iteratively solve the integer program.
- **DAL** (Discriminative Active Learning): our method, detailed above.

### 5.4 RESULTS

In figure 1 and 2 we compare the different models on the MNIST and CIFAR-10 datasets, over a large range of query batch sizes.

A clear initial observation from the experiments is that, aside from EGL, all algorithms consistently outperform random sampling on a wide range of query batch sizes. For Uncertainty Sampling, this comes in contrast to past results in (Ducoffe & Precioso, 2018) and (Sener & Savarese, 2018), which show uncertainty-based methods perform on par or worse than random sampling. Those results were attributed to the batch-mode setting, causing the examples sampled by uncertainty sampling to be highly correlated. However, our experiments show consistently that this isn't the case even for very large batch sizes. For (Ducoffe & Precioso, 2018), after reviewing the published code we believe this is due to an implementation error of the max-entropy acquisition function. As for (Sener & Savarese, 2018), we are unable to offer a good explanation for the discrepancy in our result since their implementation of uncertainty-based methods was not published. As for EGL, our result is consistent with (Ducoffe & Precioso, 2018) and an inspection of the queries made by EGL shows each query has a very high bias towards examples from a single class. A possible explanation for the discrepancy between these results and the strong results of EGL in (Huang et al., 2016) may be the architecture used for the different tasks. EGL was shown to perform well on text and speech tasks, which use recurrent architectures, while our implementation uses convolutional architectures. Since EGL uses the gradient with respect to the model parameters as the score function, it is quite plausible that the architecture used can have a big effect on the performance of the method.

The next observation to be made is that once the batch size is large enough, all of the baseline methods are quite on par with each other, with results within the statistical margin of error. When the query batch size is 100 we see a clear distinction between the performance of the uncertainty based methods and the performance of DAL and Core-Set. As the batch size increases to 1000, the distinction still exists but DAL outperforms Core Set. However, when we observe the results for CIFAR-10, a more realistic dataset with larger query batch sizes, we see all of the methods having essentially the same performance with a slight advantage for our method in the batch size of 10000.

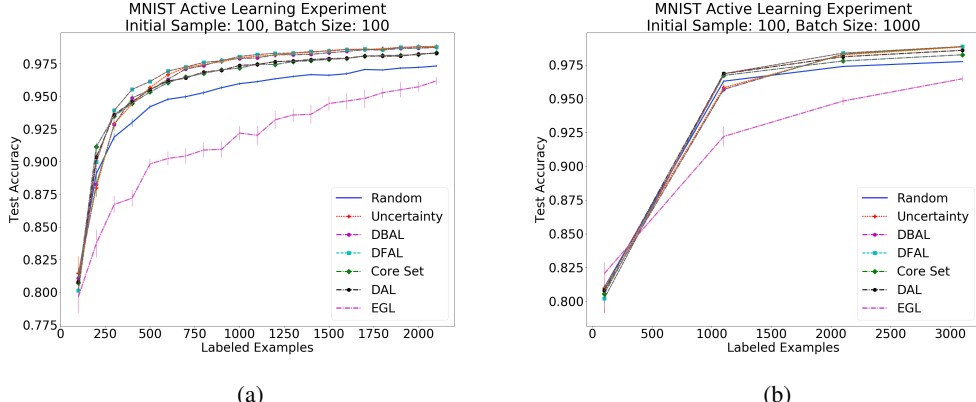

Figure 1: Accuracy plots of the different algorithms for the MNIST active learning experiments. The results are averaged over 10 experiments and the error bars are empirical standard deviations. (a) Query batch size of 100. (b) Query batch size of 1000.

## 5.5 RANKING COMPARISON

Following (Huang et al., 2016), we compare the rankings of the different algorithms on the MNIST unlabeled dataset in the first iteration of the experiment. For pairs of algorithms, we plot the rankings in a 2D ranking-vs-ranking coordinate system, such that a plot close to the diagonal implies the two algorithms score the unlabeled dataset in a similar way.

Figure 3 compares Uncertainty, DBAL, DFAL and DAL. Core-Set is not compared since it does not output a ranking over the unlabeled set, but rather only outputs a query batch.

Unsurprisingly, we observe that Uncertainty and DBAL are similar since their acquisition function is the same and only the confidence measures slightly changed. On the other hand, while DFAL uses a very different method for ranking, we see that it ends up ranking the unlabeled set in a very

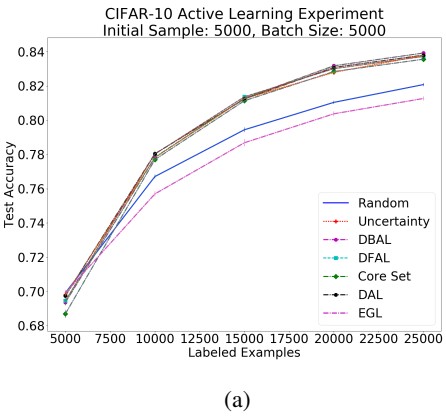 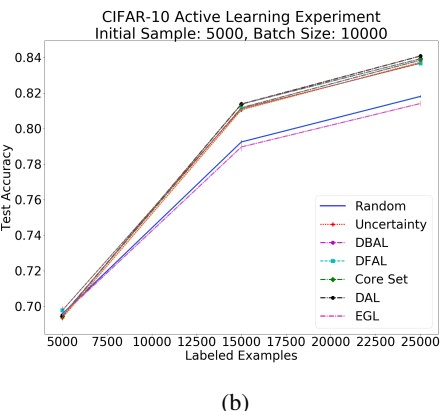

(a)          (b)

Figure 2: Accuracy plots of the different algorithms for the CIFAR-10 active learning experiments. The results are averaged over 20 experiments and the error bars are empirical standard deviations. (a) Query batch size of 5000. (b) Query batch size of 10000.

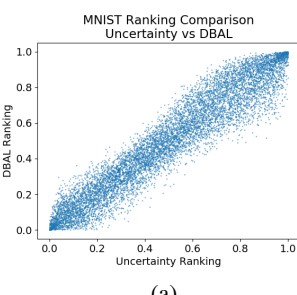 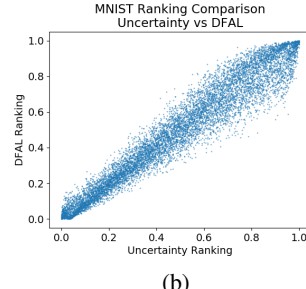 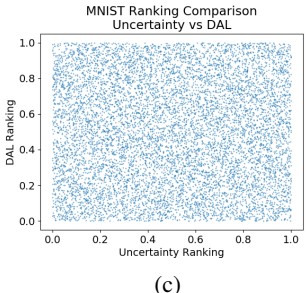

(a)          (b)          (c)

Figure 3: Comparing the ranking of the MNIST dataset according to the score of different algorithms. (a) Uncertainty vs DBAL. (b) Uncertainty vs DFAL. (c) Uncertainty vs DAL.

similar way to DBAL and Uncertainty, suggesting that even in a very non-convex function like neural networks, a margin-based approach behaves similarly to an uncertainty-based one.

When comparing DAL to the other methods however, we see a big difference in the ranking which is completely uncorrelated with the rankings of the other methods. This suggests that DAL selects unlabeled examples in a completely different way than the baseline methods, due to the formulation of the active learning objective as a binary classification problem.

## 6 CONCLUSION

In this paper we propose a new active learning algorithm, DAL, which poses the active learning objective as a binary classification problem and attempts to make the labeled set indistinguishable from the unlabeled pool. We empirically show a strong similarity between the uncertainty and margin based methods in use today, and show our method to be completely different while obtaining results on par with the state of the art methods for image classification on medium and large query batch sizes. In addition, our method is simple to implement and can easily be extended to other domains, and so we see it as a welcome addition to the arsenal of methods in use today.

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
