# OpenReview forum: "Discriminative Active Learning"
_ICLR.cc/2019/Conference_

### Official Review · AnonReviewer2 · 2018-11-02
**Method motivation and experimental results are not very convincing**

**Rating:** 4
**Confidence:** 4

**Review:**

This paper presents a new approach to an active learning problem where the idea is to train a classifier to distinguish labeled and unlabeled datapoints and select those that look the most like unlabeled.

The paper is clearly written and easy to follow. The idea is quite novel and evokes interesting thoughts. I appreciated that the authors provide links and connections to other problems. Another positive aspect is that evaluation methodology is quite sound and includes comparison to many recent algorithms for AL with neural networks. The analysis of Section 5.5 is quite interesting.
However, I have a few concerns regarding the methodology. First of all, I am not completely convinced by the fact that selecting the samples that resemble the most unlabeled data is beneficial for the classifier. It seem that in this case just the data from under-explored regions will be selected at every new iteration. If this is the purpose, some simpler methods, for example, relying on density sampling, can be used. Could you elaborate how you method would compare to them? I can see this method as a way to measure the representativeness of datapoints, but I would see it as a component of AL, not an AL alone. What would happen it is combined with Uncertainty and you use it to labeled the points that are both uncertain and resemble unlabeled data?
Besides, the proposed approach does not take the advantage of all the information that is available to AL, in particular, it does not use at the information about labels. I believe that labels contain a lot of useful information for making an informed selection decision and ignoring it when it is available is not rational.
Next, I have conceptual difficulties understanding what would happen to a classifier at next iteration when it is trained on the data that was determined by the previous classifier. Seems that the training data is non-iid and might cause some strange bias. In addition to this, it sounds a bit strange to use classification where overfitting is acceptable.
Finally, the results of the experimental evaluation do not demonstrate a significant advantage of the proposed method and thus it is unclear is there is a benefit of using this method in practice.

Questions:
- Could you elaborate why DAL strategy does not end up doing just random sampling?
- Nothing restrict DAL from being applied with classifiers other than neural networks and smaller problems. How do you think DAL would work on simpler datasets and classifiers?
- How does the classifier (that distinguished between labeled and unlabeled data) deal with very unbalanced classes? I suppose that normally unlabeled set is much bigger than labeled. What does 98% accuracy mean in this case?
- How many experiments were run to produce each figure? Are error bars of most experiments so small that are almost invisible?

Small comments:
- I think in many cases citep command should be used instead of cite.
- Can you explain more about the paragraph 3 of related work where you say that uncertainty-based approach would be different from margin-based approach if the classifier is neural network?
- Last sentence before 3.1: how do you guarantee in this case that the selected examples are not similar to each other (that was mentioned as a limitation for batch uncertainty selection, last paragraph on page 1)?
- It was hard to understand the beginning of 5.5, at first it sounds like the ranking of methods is going to be analysed.
- I am not sure "discriminative" is a good name for this algorithm. It suggested that is it opposite to "generative" (query synthesis?), but then all AL that rank datapoints with some scoring function are "discriminative".

---

> ### Author Response · Authors · 2018-11-08
> **Clarifications**
>
>  Thank you for your detailed comments and questions regarding our paper and the time and effort spent reviewing it.
>
> You are correct in saying that DAL in essence chooses the under-represented regions of the data and acts similarly to a density based approach (this can be seen in experiments on toy data as well). Indeed, for low-dimensional data there is no reason to choose to use DAL over classical density-based approaches which directly estimate the density. DAL is relevant for high dimensional data, where density-based approaches break down due to the difficulty of estimating probability densities in high dimensions. While estimating densities is difficult, we have great tools for binary classification in high dimensional data and so DAL remains effective in these domains. We briefly mention this in the first paragraph of section 3.
> As for having DAL be supplementary to uncertainty based methods, we agree that such a combination could be beneficial and improve the results of both DAL and uncertainty separately, but combining the methods in a non-naive way isn't straightforward and was outside the scope of the current work.
>
> Regarding the use of information about the labels, we would only comment that the labels aren't completely ignored in the entire active learning process of DAL. While it is true that DAL only uses the representation to pick the next sample to label, the representation itself is learned using a classifier which takes the labels of the labeled set into account. This is important, since running DAL on the raw data gives results which are much worse. The same can be said regarding the Core-Set approach, where it ignores the labels of the representation for choosing the examples to label, but uses a representation learned using the labels.
>
> As for the experimental results, the point is well taken that the results aren't convincing enough to choose DAL over the other methods in practice. Even worse, our results show that for the benchmarks of MNIST and CIFAR-10, there isn't enough information for a practitioner to decide about any of the existing methods. This speaks to a general issue in benchmarking modern AL methods, which we admittedly did not address in our paper fully by exploring other, larger datasets. Exploring the problem of benchmarking AL methods is important, but tangential to a paper that aims to introduce a new method.
> Still, we feel that our method is interesting and has merit even if it only performs comparably to the existing methods and does not improve upon them.

---

> > ### Author Response · Authors · 2018-11-08
> > **Further clarifications**
> >
> > In response to your specific questions:
> >
> > To explain the difference to random sampling, we refer you to our reply to reviewer #2, where we give an example using a finite sample space of size n where P(x) is multinomial. We explain that our choice of sample to label in this scenario is more reasonable than random sampling. We also show that form a mode-covering perspective, DAL needs n samples to cover all the n modes of the distribution while random sampling needs ~nlog(n).
> >
> > As for simpler classifiers, we used DAL with a RBF-kernel SVM on the MNIST benchmark. This gave us results better than random, but not as impressive as the neural network results. Also, we explored DAL on different synthetic 2D datasets with linear classifiers.
> > In the linear classifier setting DAL has some issues which stem from the low expressiveness of the linear classifier. DAL fits a linear classifier, trying to separate the labeled set from the unlabeled set, and chooses the example which is farthest away from the linear decision boundary in the direction of P(x~U). This is problematic because the labeled and unlabeled sets quickly become difficult to separate linearly, even if the real distribution can be linearly separated according to the labels. Secondly, the fact that we choose the farthest example from the linear decision boundary means that the method is very sensitive to outliers. Neural networks don't experience these problems as much, because they are very expressive and can take into account the density between the labeled and unlabeled examples, and so are also less sensitive to outliers.
> >
> > As for the unbalanced classes question, we chose to deal with it in a very simple way which was both weighting the classes in the loss function according to their proportions, sub-sampling the unlabeled set to be 10 times the size of the labeled set and using large batch sizes during training. For this specific scenario, 98% accuracy worked for us. This wasn't clear enough in the paper and so we added a short clarification to this in the revised version in section 4.3.
> > We agree that if one wants to incorporate the entire unlabeled set when the labeled set is small, more care needs to be given to dealing with the unbalanced class problem. In this case, we would try training the network while forcing the batches to always contain a certain percentage of examples from the labeled set.
> >
> > Regarding the experiments, we ran 20 experiments for the MNIST and CIFAR-10 benchmarks, so most error-bars are indeed too small to see.
> >
> > To answer you final comments:
> >
> > When we say that uncertainty-based approaches and margin-based approaches differ when we move away from linear binary classifiers, this is because uncertainty-based methods use the *output* probabilities of the model to decide which example to label, while margin-based methods use the euclidean distance of the *input* from the decision boundary. When the classifier is linear there is a direct, monotonous connection between the euclidean distance from the decision boundary and the output probabilities of the linear model, and so the two approaches become one and the same (the closer you are to the decision boundary, the more the model is uncertain). On the other hand, when the model is highly non-linear, the euclidean distance between the input and the decision boundary doesn't necessarily correlate to the uncertainty in the output probabilities, and this is where the two approaches can differ in their queries.
> >
> > Finally, we cannot guarantee that our queries are diverse, but we can raise the chances of that happening by using the "mini-query" trick we introduced in the paper. We simply make several successive sub-queries of smaller size, updating the binary labels for DAL after every sub-query. This makes it less likely that two similar samples will be queried in consecutive sub-queries, and makes DAL a little bit more "batch-aware". We of course pay a price in runtime the more sub-queries we use, since we need to train the binary classifier between every two sub-queries.

---

> > > ### Comment · AnonReviewer2 · 2018-11-24
> > > **Thank you for the answers**
> > >
> > > Thanks for you your answers to my comments and questions as well as the update of the article. I have read them in details.

---

### Official Review · AnonReviewer3 · 2018-11-02
**Interesting but counter-intuitive idea which works well in practice. It needs a better motivation/explanation.**

**Rating:** 6
**Confidence:** 4

**Review:**

The paper is proposing a distribution matching as a metric for active learning. Basic intuition is: if we can make the distribution of labelled and unlabelled examples similar to each other, training error in one will approximate the training error in the other. Hence, a model learned using labelled ones will do well in unlabelled ones. The main tool to enforce this distributional distance is using adversarial learning similar to GANs or gradient reversal network for domain adaptation.

The idea is definitely interesting. I am not sure about why should it work (I explained in detail later), but it does work well empirically. Moreover, it is very easy to implement. Given any learned or hard-coded features, learning a simple binary classifier is sufficient to implement the method. The mini-queries idea in 4.1 is especially interesting. Handling large batches in active learning is always a problem but this neat trick make it much easier.

I think the proposed method is counter intuitive as the discussion does not explain why should it work better than random sampling. Clearly if labelled samples are randomly sampled, labelled and unlabelled data is coming from the exactly same distribution. Hence, the distance (H-divergence, TV-distance etc.) between them is 0. My main question to authors is why does this method work better than random sampling? A similar question is; since they are coming from the exact same distribution, what is the meaning of minimizing empirical H-divergence? I think a more detailed study on a toy problem could potentially explain this. Authors can generate 1-D or 2D samples from a well defined distribution (eg. Gaussians with different means/variances for each class) and visualize what is the algorithm actually doing.

Considering my point that these data points are actually coming from the same distribution, discussion in Section 3.1 is rather unjustified. Most of the entities discussed in that section are probabilistic entities (generally speaking expected values) and does not differ between labelled and unlabelled case since they have same underlying distribution. Their empirical values are different but this is beyond the study of Ben-David(2010). Therefore, I am not sure does the Section 3.1 is contributing to the paper without any explicit connection to the empirical divergence minimization. More importantly a much similar work from domain adaptation is [Unsupervised domain adaptation by backpropagation, ICML 2015] and it should also be discussed in the paper.

Some minor issues:
- Are the hyper-parameters kept fixed for all experiments. In other words, does the training size of 5k and 15k share hyperparameters? Which might be sub-optimal.
- The experiments use very large batch sizes. A smaller batch sizes might separate the algorithms better.
- References in the text have some issues. There are missing commas between references in the text. There are also some cases where \citep should have been used but \citet is used. A careful pass over them might be beneficial.

In summary, I think the paper is interesting, easy to implement and possibly useful to the large part of the community since active learning is very important problem. I think the major weakness of the paper is the fact that authors did not give a clear explain why does it actually work. I think it is crucial for authors to provide a theoretical or an empirical study which answer this question.

---

> ### Author Response · Authors · 2018-11-08
> **Relevance of domain adaptation and clarifications**
>
>  Thank you for your thoughtful comments regarding our paper.
>
> We'll first address your comments about using the H-divergence as a motivation for our method and the expectation that DAL shouldn't improve upon random sampling. We agree that when considering the underlying distributions of the labeled and unlabeled sets, the TV distance between them is zero and that random sampling allows claiming that they are indeed the same distribution. That being said, we work under the assumption that the labeled set is very small compared to the unlabeled pool, and so it can be viewed as coming from a different, empirical distribution (for instance, one which contains only some of the modes of the original distribution). When represented in this way, for a fixed labeled set, we claim that we have a distribution shift that we must correct for using active learning. We could either sample an example randomly from the real distribution, which will give us in expectation a good solution (assuming we sample enough examples), or actively choose the examples which minimize this distribution drift which is caused by the small size of the labeled set. So, while in expectation over large labeled sets we should expect the domain shift to be zero, in essence when we are dealing with small labeled sets it is more accurate to treat the distributions as different and attempt to minimize their distance. This is mentioned shortly in the second paragraph of section 3.1.
> A simple example to motivate our reasoning can be to look at a finite sample space with n possible values for x, and P(x) being some multinomial distribution over the values of x. We also have an estimate of the probabilities, P*(x), which comes from the relative frequency of every x in out labeled set. These two distributions are different and we want to make them more similar. Our method will choose the sample to label which maximizes P(x)/(P(x)+P*(x)), which is the one where P(x)/P*(x) is largest. This is a good heuristic as opposed to picking randomly, which might even push the distributions farther apart.
> We can also motivate this in a similar way to the Core-Set approach, in which we care about representing all of the modes of the distribution (having all possible examples of x in our labeled set). If we use DAL for the above example, assuming P(x) is non zero for every x, then we will represent all of the modes after n samples. Contrary to DAL, random sampling will capture all of the modes in ~nlog(n) samples, which is a big difference if n is large and we want to have as few labeled samples as we can.
> Finally, we did run experiments on toy data when trying to compare our method to the Core-Set approach. We used two Gaussians, one having a large variance and the other having a smaller one. While the Core-Set approach favored labeling samples from the large variance Gaussian, DAL labeled samples more equally from the two Gaussians. We felt this experiment, along with other ones we ran on low dimensional data, didn't add enough to justify getting into the final paper under the ICLR page constraints, and chose to let the experiments on more realistic data be the main message.
>
> As for your comment about the similar work from domain adaptation, we thank you for bringing this paper to our attention. We discuss it briefly in section 3.1.1 of the revised submission.
>
> As for the minor issues you raise:
> The hyper-parameters are indeed kept fixed, and we agree this could be sub-optimal. Using cross validation in every iteration of the active learning experiment for every AL method for the amount of experiments we ran was computationally too expensive for us. Still, since all algorithms were tested in the same playing field with the same parameters, we think our results can be trusted.
> The batch sizes were indeed quite large, as we are addressing active learning for neural networks where using small batch sizes is often impractical since we are trying to amass a relatively large dataset. We agree that in the domain of small batch sizes there could be stronger differences between the methods, specifically a bigger advantage to uncertainty based methods compared to DAL and the Core-Set approach.

---

> > ### Comment · AnonReviewer3 · 2018-11-21
> > **Theoretical motivation is still not clear.**
> >
> > Thanks for the update. I think it is more intuitive right now when you think in terms of empirical distributions. Hence, it largely addresses my issue with the counter intuitiveness. However, the text of the paper is still talking about the data distribution (or population distribution). Even more importantly, entire theory is about data/population distribution. Hence, I the lemmas stated in Section 3.1 is neither helpful nor correct in empirical data case. I would strongly suggest either re-stating them for the empirical distribution case or removing them completely. It is too confusing in this form. This is my only major concern left for the paper.

---

> > > ### Author Response · Authors · 2018-11-22
> > > **Revision of section 3.1**
> > >
> > > Thanks for your comments regarding section 3.1. Following your comments, we agree that the definitions and lemmas as presented are not relevant to our method and the relation to domain adaptation should have been presented differently.
> > > We have revised the section to better reflect our empirical distribution setting, while maintaining the relation to domain adaptation which we feel is important to mention. This includes restating the definitions to reflect the empirical distributions L and U and removing the unnecessary lemmas and definitions which made the section too cumbersome.
> > >
> > > We hope these changes answer your concerns.

---

> > > > ### Comment · AnonReviewer3 · 2018-11-26
> > > > **Thanks for the updates**
> > > >
> > > > Thanks for the updates. I think the methodology and intuition is much clearer. Considering the promising results and the interesting idea, I believe this paper can be an interesting addition to ICLR. I think the paper needs some re-writing since it changed significantly during rebuttal period. Still there are some references to parts which have been removed/changed. I believe this minor re-write can be done before the camera ready deadline if it is accepted. I am increasing my score to 6.

---

### Official Review · AnonReviewer1 · 2018-11-02
**very nice paper with neat idea, good theoretical intuition/justification, clear transparent code/algorithm, and good experimental results**

**Rating:** 8
**Confidence:** 4

**Review:**

Thank you for this enjoyable paper.

Summary: The authors propose a novel approach to active learning as follows. At each iteration they develop a classifier that can discriminate between the samples in the labeled and unlabeled sets; they select the top few samples that are most likely to be from the unlabeled set as per this classifier, and request the oracle to provide labels for this batch next. This simple idea is shown to have a principled basis and theoretical background, related to GANs and to previous results from the literature. They provide clear algorithms and open source code for easy verification, and public testing. They provide good experimental verification on CIFAR-10 and MNIST benchmarks. I personally look at new papers more for novel ideas and good intuition/theoretical justification than an immediate improvement in benchmark results, so I enjoyed this paper thoroughly.

Results: Among other things they show that their algorithms ranks the samples to be next labeled quite differently than uncertainty sampling based approaches; that their method is at least as accurate/sample-efficient as the state of the art ; and that some previously published experimental results are incorrect(!). As the authors will probably agree I am not convinced the proposed method is better than previous algorithms in any statistically significant way, but the novel idea itself is worth publishing even if it is just as good as the state of the art.

Novelty: I liked the paper very much because it provides quite an innovative new approach to look at active learning, which resembles GANs and Core set ideas in some ways, yet differs in significant ways that are critical for active learning. I've been working and publishing in related areas for a long time so I genuinely found your central idea refreshing and new.

Relevance: The paper is very relevant to the ICLR community and addresses critical questions.

Question:
My intuition as a Bayesian is that we most need to find labels that maximize the mutual information I(y,w) where w are the weights of the neural net. In practice this corresponds to the samples x which have the maximum class uncertainty, but for which the parameters under the posterior disagree about the outcome the most, eg see discussion below equation 2 for  Bayesian Active Learning by Disagreement (BALD) in this paper https://arxiv.org/pdf/1112.5745.pdf . In essence: The above means that the labels that provide most information about the classification model are most valuable for active learning.

However, your approach intuitively ignores the conditional distribution(ie py(|x)), and instead tries to make the original unconditional distribution p(x) between the labeled and unlabeled sets similar. Yet, it works beautifully. So: Why does this work? What is the intuition?

---

> ### Author Response · Authors · 2018-11-08
> **Some intuition for ignoring P(y|x)**
>
>  We appreciate the time and effort of reviewing our paper, and thank the reviewer for the kind words.
>
> As for an intuition for why our method should work while ignoring the conditional probability P(y|x), we would not view the problem we are trying to solve as one which is tailored only to classification, and so we do not need access to the probability over the labels. As you say, we are trying to find a subset of the distribution which represents the true distribution as much as possible. Under the assumption that the distribution of labels is strongly dependent on the inputs (a reasonable assumption in our view), then getting a labeled set which correctly captures P(x) should reasonably capture P(x,y) as well, which is what we need for classification. A similar notion can be found in the Core-Set approach, where Lipschitz assumptions on the labeling function allows for a bound on the test loss of the classifier, and so the objective becomes to cover the representation of the data (which ignores the labels as well).
> We would also note that while we do ignore P(y|x) in DAL, there is still information from the labels that is used during the entire active learning process, since we are running DAL on a representation learned by a classifier that uses the labels in the labeled set.
> Having said all of that, it is quite reasonable to think that a method which combines DAL with other methods which do use the label information could improve on our results.

---

### Public Comment · (anonymous) · 2018-11-07
**A question on the domain adaptation bound**

In Lemma 3.1 of Ben-David (2010) adaptation bound, the gap between the empirical and expected diveregence depends on the  VC dim of the classifier. However, in the deep neural network such a value will be **extremely** high. Therefore when you only minimize the empirical divergence, this Lemma can not support it will have a small divergence between the two **real** distributions, since the complexity term will be much bigger than the empirical divergence.

---

> ### Author Response · Authors · 2018-11-08
> **True - we only use the bound as motivation**
>
> Thank you for your interest and question regarding our paper.
> You are correct in that the bound we present is very far from being tight when the hypothesis class in question is a family of neural networks. Unfortunately, this problem exists in all generalization bounds that use a capacity measure of the hypothesis class to derive the bound.
> For this reason, we treat the theoretical bound from domain adaptation as merely a motivation for our algorithm.

---

### Author Response · Authors · 2018-11-08
**Summary of changes to the revised version of the paper**

We thank the reviewers and others for the effort they spent in reviewing our paper. We hope our responses help answer some of the reviewer concerns. We have uploaded a revised version of our paper to answer some of these concerns. We tried to be concise as to not exceed the 8 page restriction too much. The changes are as following:
1. We added a discussion comparing our work to the unsupervised domain adaptation paper mentioned by reviewer #2. This appears section 3.1.1.
2. We clarified section 4.3 which deals with handling overfitting issues. We explain that the 98% accuracy we mention is relevant for an unlabeled pool that is ten times the size of the labeled set.
3. We changed the citations to be /citep instead of /citet and added commas between consecutive citations.

---

> ### Author Response · Authors · 2018-11-22
> **Revision of section 3.1**
>
> We have uploaded a second revision to the paper, following the comments of reviewer #3.
> Section 3.1, which deals with the relation between DAL and domain adaptation, has been revised by removing unnecessary definitions and lemmas, and restating other definitions to better reflect the empirical distribution setting of DAL.
> We hope these changes make the paper more accurate and clear, and thank reviewer #3 for the comments.

---

### Public Comment · ~Mélanie_Ducoffe1 · 2018-11-13
**DFAL's implementation**

Tahnk you for this paper. I do agree that domain adaptation is highly relevant for the active learning community, and thus your method opens up promising results for the field.

As one of the main author of DFAL, I would like to point out a misinterpretation of our method in your implementation: DFAL queries the data with the smallest adversarial perturbation, but adds also the corresponding adversarial examples to the labeled training set, using twice the same label. In our experiments (check our arxiv version), we noticed that adding the adversarial samples in the labeled training set do increase the test accuracy.

On the other hand, I disagree with your overall statement “none of the state-of-the-art methods of today are clearly better than uncertainty sampling”. DFAL is a top score approach, thus it does not take into account the correlations of the samples, as any batch mode active learning methods do. This is why we conduct our experiments on small query batch sizes (10 in most of our experiments). Indeed, in Figure 3 of your paper, the highest ranking samples of DBAL/DFAL are not correlated to the most uncertainty samples. Then by using a smaller size of queries, you should notice some improvements of those methods.
For fairness sake, it would be better either to run your experiments with smaller query batch sizes or use a diversity-based criterion with DFAL, and other top score approaches such as DBAL or EGL. For example, [Submodularity in Data Subset Selection and Active Learning, Wei et al. 2015] has been combined with uncertainty selection on deep neural networks.

In case the authors wish to extend their experiments to shallow classifiers, I would highly suggest a comparison with [Querying Representative And Discriminative Batch Mode Active Learning, Wang et al. 2013], that combines uncertainty selection and MMD discrepancy to cover at best the distribution.

I just intend this as feedbacks for revision, I don't intend for this to have any bearing on whether the paper is accepted or rejected.

---

> ### Author Response · Authors · 2018-11-15
> **Response**
>
> Thank you for your comments and for your original DFAL paper, which we found to be very compelling.
>
> We apologize for misinterpreting your method in that we did not add the adversarial examples to our labeled set along with the original samples, which could have increased performance as you wrote. However, this sort of adversarial training is known to be a good regularizer for the training process in general and so we would say that any active learning method can benefit from this sort of data augmentation. If we understand your method correctly, the choice of example to label and the adversarial data augmentation can be decoupled, and this way all methods may benefit from it. So, adding the adversarial examples to the training set only for one AL method could be seen as an unfair advantage to that method.
>
> Regarding our statement that no method clearly beats uncertainty sampling, we would clarify that this is definitely true for large batch sizes (100 and more), which we claim are the only viable query sizes for deep learning applications. This means that a deep learning practitioner when faced with the need to choose an active learning method will have a hard time convincing himself to use a method that is more complicated than uncertainty sampling, given the current empirical evidence (we unfortunately have to include DAL in this assessment). We feel this is an important point to make, as it contradicts the results of the core set paper, for example.
> Even though we feel the smaller query batch size domain isn't as relevant to deep learning applications, we reran our experiments again on the MNIST dataset with an initial labeled set of size 100 and query batch size of 10 as you suggested. These experiments do show a slight advantage to DFAL over the other methods, but one that is small compared to the one presented in your paper. The graphs look quite similar to the query size of 100 in our paper - DFAL wins initially (up to ~400 labeled samples), then DBAL and uncertainty sampling begin to catch up to it. The clear difference between the DAL and Core Set methods and DFAL, DBAL and US is present as in the query size 100 experiments in our paper, but surprisingly we see that DAL clearly beats the Core Set approach, which turns out not to outperform random sampling. This is different from your results for Core Set (which are close to DFAL), but we would need more time to understand what we did differently and so we reserve judgement about Core Set at this point.
> Given these results, we would still say that there needs to be more empirical evidence in order to choose DFAL or any other method over uncertainty sampling in small query sizes.

---

### Meta-Review · Area_Chair1 · 2018-12-13
**Novel active learning approach, but work could still benefit from revisions and additional baselines**

**Confidence:** 3
**Recommendation:** Reject

**Metareview:**

This paper proposes a novel and interesting active learning approach, that  trains a classifier to discriminate between the examples in the labeled and unlabeled data at each iteration. The top few samples that are most likely to be from the unlabeled set as per this classifier are selected to be labeled by an oracle, and are moved to the labeled training examples bin in the next iteration. The idea is simple and clear and is shown to have a principled basis and theoretical background, related to GANs and to previous results from the literature. Experiments performed on CIFAR-10 and MNIST benchmarks demonstrate good results in comparison to baselines.
During the review period, authors considered most of the suggestions by the reviewers and updated the paper. Although the proposed method is similar to density-based active learning methods, as also suggested by the reviewers, baselines do not include such approaches in the comparison experiments.